# The Impact of the Global Pandemic on Veterans with Serious Mental Illness (SMI): Healthcare Utilization and Mortality

**DOI:** 10.3390/bs14050356

**Published:** 2024-04-24

**Authors:** Isabella Soreca, Monique Boudreaux-Kelly, Yeon-Jung Seo, Gretchen Haas

**Affiliations:** 1VISN 4 Mental Illness Research, Education and Clinical Center, VA Pittsburgh Healthcare System, Pittsburgh, PA 15240, USA; 2Department of Statistics, University of Pittsburgh, Pittsburgh, PA 15213, USA; yes18@pitt.edu; 3Department of Psychiatry and Psychology, University of Pittsburgh, Pittsburgh, PA 15213, USA

**Keywords:** serious mental illness, mortality, healthcare utilization, COVID-19

## Abstract

Background: Individuals with serious mental illness (SMI) experience barriers to accessing and engaging with healthcare, which may have been exacerbated during the emergence of the global pandemic and the rapid shift to telemedicine platforms, substantially decreasing healthcare utilization for non-COVID-19 disorders. Important repercussions on morbidity and mortality may be seen in the months and years to come, which may disproportionately affect high-risk populations, such as patients with SMI, with reduced access to technology platforms. In this study, we explored the impact of the pandemic on healthcare utilization and all-cause mortality rate in SMI compared to non-SMI individuals for the months of March–September 2020 and the same two quarters in 2019. Methods: Data were obtained from the VA Corporate Data Warehouse (CDW), a data repository from clinical and administrative VA systems. The sample included veterans with ≥1 outpatient clinical encounter nationally between 1 January 2019 and 31 December 2020. Results: The cohort for this study included 1,018,047 veterans receiving care through the Veterans Health Administration between 2019 and 2020. Of those, 339,349 had a diagnosis of SMI. Patients with SMI had a significantly larger pre–post-pandemic decrease in outpatient (49.7%, *p* < 0.001), inpatient (14.4%, *p* < 0.001), and ED (14.5%, *p* < 0.001) visits compared to non-SMI patients. Overall, 3752 (1.59%) veterans without SMI and 4562 (1.93%) veterans with SMI died during our observation period. Veterans without SMI who died during the observation period were more likely to have had a positive COVID-19 test compared to veterans with SMI. Unadjusted analyses showed that veterans with SMI were approximately 2.5 times more likely to die than veterans without SMI during the first 6 months of the pandemic, compared to the same two quarters of the previous year. However, after adjustment by pertinent covariates, the predictors associated with an increased risk of death from SMI were older age, being male, a higher CAN score, more inpatient stays in the pre period compared to post, and a positive COVID-19 test. Discussion: Consistent with our initial hypothesis, all the indices of healthcare utilization, namely the number of outpatient, inpatient, and ED visits, significantly decreased between pre- and post-pandemic and did more so for veterans with SMI, despite having more chronic medical illnesses and being prescribed more medications than veterans without SMI. On the other hand, while mortality was greater post-pandemic, factors such as age, morbidity, and having a positive COVID-19 test predicted mortality above and beyond having an SMI diagnosis.

## 1. Introduction

Approximately 4–5% of veterans receiving healthcare through VA are diagnosed with a serious mental illness (SMI) [1], such as psychotic spectrum disorders, resulting in functional impairment in multiple areas. This small percentage of the population experiences poorer medical outcomes [2], increased mortality [3], and increased healthcare spending [4]. Yet, individuals with SMI experience barriers to accessing healthcare, and physical and mental health complaints often go untreated or unrecognized [5]. The emergence of the global pandemic and the declaration of the state of emergency has caused a rapid shift in the way healthcare is delivered, with most preventative and routine follow-up care for chronic disorders, including SMI, being transitioned to online platforms [6]. While telemedicine has been an invaluable tool under these circumstances, allowing us to maintain access to care, data from the general population suggest that healthcare utilization for non-COVID-19 disorders has substantially decreased when compared to the utilization pattern during the first six months of the prior year(s). This includes, as expected, the care and management of chronic diseases and preventative measures [7,8], as well as reduced access to potentially life-saving interventions for acute illnesses [9,10].

This acute decline in routine care may have important repercussions on morbidity and mortality in the months and years to come, which may disproportionately affect high-risk populations, such as patients with SMI, with reduced access to technology platforms [11]. This study will address these gaps in research by investigating the patterns of healthcare utilization and mortality prior to and during COVID-19.

In this study, we explored the impact of the pandemic on healthcare utilization in SMI compared to non-SMI individuals. Specifically, we assessed the number of outpatient, ER, and inpatient visits to healthcare specialists for the months of March–September 2020 and compared it the same two quarters for 2019 in SMI and non-SMI individuals.

We also assessed the mortality rate for the six months following the state of emergency declaration in the US (i.e., 15 March–15 September 2020) and compared it to the mortality rate during the same period of 2019 in SMI and non-SMI individuals.

## 2. Material and Methods

Data were extracted from the VA Corporate Data Warehouse (CDW), a repository of clinical and administrative data from the VA systems. The cohort comprised veterans with ≥1 outpatient clinical encounter nationally between 1 January 2019 and 31 December 2020. This study was approved by the VA Pittsburgh Healthcare System (VAPHS) Institutional Review Board.

SMI cohort: Details about the cohort selection were detailed in a previous paper [12]. Briefly, the cohort comprised veterans from any facility nationally, who had ≥1 outpatient visits from any VA facility during the calendar years of 2019–2020. We then assessed the cohort for SMI diagnosis, retaining as valid diagnoses only when a patient had one or more of the ICD codes of interest during the calendar years of 2019–2020. When veterans had multiple SMI diagnoses, we selected the diagnosis that had the most recurrences. Non-SMI controls were matched by age, gender, race, and ethnicity. For every patient with SMI, we included two matching controls.

SMI diagnoses: The SMI sample included individuals with the following diagnoses: bipolar disorder (ICD-10 codes F30-31), schizophrenia (ICD-10 codes F20, F22-24), schizoaffective (ICD-10 code F25), and psychosis (ICD-10 codes F28-29).

Healthcare utilization and mortality: The number of outpatient, inpatient, and ER visits for any diagnostic code (medical or psychiatric) was used as an index of healthcare utilization.

Mortality: The vital status was used as an index of all-cause mortality.

Sociodemographic variables: We included age, sex, race, ethnicity, and region per the data available in the Electronic Health Record. Age was calculated using date of birth and index date, and ages 18 to 89 were included in the cohort. Sex was male or female. Race was White, Black, Other (i.e., all other individual races: American Indian or Alaska Native; Asian; Native Hawaiian; or Other Pacific Islander), and missing. Ethnicity was Hispanic or not.

Clinical variables: The presence of comorbid conditions during the prior 2-year period were included: Obesity, Type 2 Diabetes, COPD, Congestive heart failure, Stroke, Hyperlipidemia, Depression, Anxiety, Smoking status, Illicit Drug Use, and Alcohol Use. The presence of the following medications/drug classes during the prior 2-year period was recorded: Antidiabetic, Antihypertensive, Antilipemic, Antidepressant, Anxiolytic/Sedative/Hypnotic, Antineoplastics, Antipsychotic, Opioids, Antiarrhythmics, Antiretroviral, and Thyroid. Body mass index (BMI) at index date was calculated via the formula (Weight (lb)/(Height (in) × Height (in)) × 703 by using the weight measured during pre-index that was closest to the index date and the height measured at any time in the prior 2 years. The Care Assessment Need (CAN) score [13] was used to assess the likelihood of hospitalization or death at 1 year and 90 days using the most recent score within the 2-year pre-index date. CAN is a risk stratification tool developed by the Veterans Health Administration to aid primary care providers in patient management and care coordination by making veteran-specific probabilistic predictions of future adverse events. The CAN score is generated from VA electronic health record data that includes demographics, information on medical conditions, number of diagnoses, vital signs, medications, laboratory tests, use of care coordination resources, and overall healthcare utilization. A higher CAN score is associated with a higher risk of an adverse outcome. Utilization over the pre-index year was assessed with frequencies of visits in outpatient, inpatient, and emergency department settings.

COVID-19 testing: Veterans’ data for testing for COVID-19 were obtained from all outpatient encounters. Lab test results were categorized as positive, negative, or inconclusive. Test results were considered positive if the veteran had at least one positive COVID-19 test result anytime between 1 January 2019 and 31 December 2020.

Data Analysis. We used two-sample group comparisons to test whether patients with an SMI diagnosis would have significantly different percentages of outpatient, inpatient, and ER visits and all-cause mortality compared to patients without an SMI diagnosis and whether the pre- and post-pandemic visits number change varied by SMI status. Continuous variables measured were age, CAN score, and BMI, at index which were presented as means and standard deviations and compared using the non-parametric Wilcoxon rank-sum tests. For categorical predictors, data were summarized using proportions and odds ratios (ORs). Chi-square tests were used to assess the association between two categorical variables. The level of statistical significance was set at 0.05 (*p*  <  0.05).

Generalized linear models (GLMs) were used to examine the association between SMI and change in healthcare utilization, as well as the association of all-cause mortality in veterans with SMI, while adjusting for other potential risk factors. Age; sex; race; ethnicity; region; number of medication drug classes used; the CAN score for hospitalization at 1 year; 1-year pre-index utilizations of outpatient, inpatient, and ED services; and COVID-19 status were included as covariates in both models.

## 3. Results

The cohort selected for this study comprised 1,018,047 veterans receiving care through the Veterans Health Administration between 2019 and 2020. Of those, 339,349 had received a diagnosis of SMI. Nearly 84% were males. The details are shown in Table 1. Overall, veterans with SMI were more likely to have medical comorbid conditions and to be prescribed any of the tracked medication categories compared to veterans without SMI. SMI was associated with an approximately 2.4-fold increased odds of depression. Veterans with SMI had about 200% greater odds of being diagnosed with anxiety compared to non-SMI.

Changes in Healthcare utilization pre- and post-pandemic: The number of outpatient, inpatient, and ED visits significantly decreased between pre- and post-pandemic for both SMI and non-SMI patients. Patients with SMI had a 1.46-fold decrease in outpatient visits, a 0.27-fold decrease in inpatient visits, and a 0.23-fold decrease in ED visits between pre- and post-pandemic. In adjusted analyses, patients with SMI had a significantly larger pre–post-pandemic decrease in outpatient (50%, *p* < 0.001), inpatient (14%, *p* < 0.001), and ED (15%, *p* < 0.001) visits compared to non-SMI patients (Table 2).

The pattern of outpatient healthcare utilization by SMI diagnostic categories in adjusted and unadjusted analyses is shown in Table 3. Compared to patients without SMI, patients with bipolar disorder and with psychosis NOS had approximately 18% and 15% fewer outpatient visits, respectively, in the first two quarters post-pandemic, while patients with schizoaffective disorder had 3.65% more visits in the post-pandemic period. No difference in outpatient care utilization was found for patients with schizophrenia compared to non-SMI. Compared with non-SMI patients, patients with bipolar disorder and with psychosis NOS had approximately 17% and 5% fewer inpatient visits, respectively, in the post-pandemic period, while patients with schizophrenia had 5.53% more post- then pre-pandemic inpatient care compared to non-SMI patients. Similarly, the pattern of ED visits saw a 6.58% and 8% decrease for patients with bipolar disorder and psychosis NOS compared with non-SMI patients. No differences in ED visits between pre- and post-pandemic were observed for patients with schizophrenia and schizoaffective disorder compared to non-SMI.

Overall mortality and Changes in mortality pre- and post-pandemic: Overall, 5894 (1.74%) veterans with SMI and 4871 (0.72%) Veterans without SMI died during our observation period (Table 2). Veterans without SMI had a better chance of survival than veterans with SMI, regardless of COVID-19 status, compared to veterans with SMI (Table 4), with an odds ratio for death, adjusted for their COVID-19 test results, of 1.59 (Mantel–Haenszel 95%CI: 1.53, 1.66).

Unadjusted analyses showed that veterans with SMI were approximately 2.5 times more likely to die than veterans without SMI during the first 6 months of the pandemic compared to the same period of the previous year. However, after adjustment by pertinent covariates (Table 2), the predictors associated with increased risk of death from SMI were older age, being male, higher CAN score, more inpatient stays in the pre period compared to post, and a positive COVID-19 test (Table 5). When looking at individual SMI diagnoses in adjusted analyses, patients with bipolar were 27% less likely to die, and patients with NOS psychosis were 25.9% more likely to die than patients without SMI.

## 4. Discussion

This study explored the impact of the pandemic on healthcare utilization and all-cause mortality in individuals with and without SMI in a nationally representative sample of US military veterans. The life expectancy of the general population has been steadily increasing, yet people with SMI, including diagnoses of schizophrenia, schizoaffective disorder, and bipolar disorder, have a reduced life expectancy of 13 years for men and 12 years for women [14]. While non-natural deaths, such as suicide and accidents, account for nearly 20% of the increased mortality, most deaths can be ascribed to chronic diseases such as cardiovascular and respiratory illnesses, diabetes, cancer, and digestive disorders [15,16]. Most of these conditions are partially attributable to modifiable lifestyle factors and suboptimal healthcare use (e.g., prevention and treatment adherence) [17,18,19]. The declaration of a state of emergency in March 2020 caused a rapid shift in how healthcare was delivered in the US and elsewhere. Many “non-essential” healthcare services, which included the outpatient management of chronic illnesses, were placed on hold and/or were delivered remotely via telehealth platforms or phone. Evidence has emerged that the number of visits for preventative/routine care (for example, child immunizations) and serious and potentially life-threatening conditions (for example, ER visits for acute MI) drastically dropped during the first six months of the pandemic, with possible repercussions for non-COVID-19-related morbidity and mortality in the months/years to come. As the largest integrated healthcare system in the United States, VA health record data offer unique insights into the care provided to veterans, pointing to areas for potential improvement, including medical care of veterans with SMI. Consistent with our initial hypothesis, all the indices of healthcare utilization, namely the number of outpatient, inpatient, and ED visits, significantly decreased between the pre- and post-periods of the declaration of a global pandemic and did more so for veterans with SMI despite having more chronic medical illnesses and being prescribed more medications than veterans without SMI.

On the other hand, while the overall number of deaths was greater post- than pre-pandemic, as expected, the pattern was more nuanced than initially anticipated. The increase in overall mortality has been reported in the literature [20,21] and has been attributed to the direct effects of COVID-19 or COVID-19-related complications. While our data encompass all-cause mortality, not allowing us to determine the exact reason for the observed increased death toll during the pandemic, the adjusted analysis accounting for other covariates, including comorbidities and having a positive COVID-19 test result, suggest that the reasons for excess mortality were different for SMI and non-SMI patients. For example, non-SMI individuals who died during the observation period were more likely to have had a positive COVID-19 test compared to SMI patients. However, the excess mortality in SMI individuals in the pandemic period appears best explained by a composite set of demographic and health factors. Taken together, these data seem to suggest that, while the pandemic may have brought on decreased healthcare utilization and increased mortality for SMI and non-SMI alike, pre-existing factors may have played a larger role in the SMI death toll. The increased mortality, in the face of reduced healthcare utilization, for this group with a higher prevalence of chronic medical conditions could be directly ascribed to neglecting healthcare needs. These results extend previous work from other groups, reporting a steeper increase in all cause-mortality for patients with SMI compared to non-SMI during the pandemic [22], which stretched long after the initial period of the pandemic [23,24]. Also, of interest is the finding that different SMI diagnostic groups saw a different pattern of healthcare utilization change pre- and post-pandemic, as well as different pattern of mortality rates, which may reflect different degrees of functional impairment and medical comorbidities.

Several limitations need to be carefully considered when interpreting these results.

First, patients receiving care through the VA Healthcare System may have specific characteristics, namely demographics and comorbidities, that make these data not immediately generalizable to community samples.

Moreover, our diagnoses (medical and psychiatric) relied on the presence of specific ICD codes rather than on direct assessments via tests and structured interviews. However, while this method may reduce the reliability of each diagnostic category, it is reflective of actual clinical practice.

Finally, some veterans may receive additional care outside the VA; hence, a small subgroup of the patients included in this study may have an incomplete assessment of their SMI diagnosis and comorbidities.

## 5. Conclusions

This study underscores a differential impact of the pandemic on healthcare utilization and all-cause mortality across a nationally representative sample of US military veterans with and without SMI, suggesting that the emergence of the global pandemic may have exacerbated the pre-existing gap in mortality. The pandemic has led to a disruption in routine care, potentially leading to delays or neglect in receiving care for acute and chronic physical conditions. These disruptions may have impacted patients with SMI to a larger extent than non-SMI individuals, as their healthcare needs are more complex and multifaceted. Policies aimed at improving access to care during a state of emergency, especially tailored for individuals with SMI, are needed.

## Figures and Tables

**Table 1 behavsci-14-00356-t001:** Sample characteristics.

Measurement	SMI (N = 339,349)	Non-SMI (N = 678,698)	OR (95% CI)	Sig **
Age *, Mean ± SD	55.05 ± 14.84	55.05 ± 14.84	1.00 (0.999, 1.001)	*p* = 0.99
Male *, N (%)	285,178 (84.04%)	570,356 (84.04%)	1.00 (0.989, 1.011)	−0.0002 (−0.0017, 0.0013); *p* = 0.79
Race *, N (%)				*p* < 0.001
White	230,150 (67.82%)	459,656 (67.73%)	Reference group	Reference group
Black	86,751 (25.56%)	172,328 (25.39%)	−1.07 (−1.080, −1.069)	*p* < 0.001 Black vs. White
Other	12,878 (3.79%)	25,508 (3.76%)	−0.74 (−0.747, −0.737)	*p* < 0.001 Other vs. White
Missing	9570 (2.82%)	21,206 (3.12%)	−0.92 (−0.926, −0.916)	*p* < 0.001 Missing vs. White
Hispanic *, N (%)	26,855 (7.91%)	53,710 (7.91%)	1.00 (0.985, 1.015)	0.0000 (−0.0011, 0.0011); *p* = 0.99
Prior 2-Year Comorbid Conditions, N (%)				
Obesity (ICD or Procedure)	58,070 (17.11%)	97,698 (14.39%)	1.23 (1.214, 1.242)	0.0272 (0.0257, 0.0287); *p* < 0.001
Type 2 Diabetes (ICD)	80,127 (23.61%)	134,159 (19.77%)	1.25 (1.242, 1.267)	0.0384 (0.0367, 0.0402); *p* < 0.001
COPD (ICD)	43,986 (12.96%)	49,322 (7.27%)	1.90 (1.875, 1.926)	0.0569 (0.0557, 0.0582); *p* < 0.001
Congestive heart failure (ICD)	14,301 (4.21%)	19,825 (2.92%)	1.46 (1.431, 1.495)	0.0129 (0.0121, 0.0137); *p* < 0.001
Stroke (ICD)	153,817 (45.33%)	279, 868 (41.24%)	1.18 (1.172, 1.191)	0.0409 (0.0389, 0.0430); *p* < 0.001
Hyperlipidemia (ICD)	150,637 (44.39%)	279,622 (41.20%)	1.14 (1.130, 1.149)	0.0319 (0.0299, 0.0339); *p* < 0.001
Depression (ICD)	126,829 (37.37%)	136,017 (20.04%)	2.38 (2.359, 2.403)	0.1733 (0.1714, 0.1752); *p* < 0.001
Anxiety (ICD)	175,300 (51.66%)	178,644 (26.32%)	2.99 (2.965, 3.017)	0.2534 (0.2514, 0.2553); *p* < 0.001
Smoke (ICD, Procedure, or HF)	116,070 (34.20%)	125,617 (18.51%)	2.29 (2.268, 2.310)	0.1570 (0.1551, 0.1558); *p* < 0.001
Illicit Drug Use (ICD)	79,457 (23.41%)	35,026 (5.16%)	5.62 (5.544, 5.694)	0.1825 (0.1810, 0.1841); *p* < 0.001
Alcohol Use (ICD)	81,719 (24.08%)	56,205 (8.28%)	3.51 (3.472,3.554)	0.1580 (0.1564, 0.1596); *p* < 0.001
CAN Comorbidity Score at Index (1–99%), Mean ± SD				
Mortality				
In 1 Year	45.24 ± 27.84	37.61 ± 25.96	1.20 (1.199, 1.207)	*p* < 0.001
In 90 Days	45.29 ± 28.00	38.72 ± 26.20	1.17 (1.166, 1.174)	*p* < 0.001
Hospitalization				
In 1 Year	68.37 ± 25.58	44.71 ± 29.59	1.53 (1.524, 1.535)	*p* < 0.001
In 90 Days	68.50 ± 25.00	45.16 ± 29.00	1.52 (1.512, 1.522)	*p* < 0.001
Body mass index at Index date, Mean ± SD	30.10 ± 6.36	30.54 ±6.17	0.99 (0.985, 0.987)	*p* < 0.001
Prior 2-Years Number of Medication Drug Classes Used, Mean ± SD	3.22 ± 2.03	1.76 ± 1.76	1.83 (1.825, 1.839)	*p* < 0.001
Prior 2-Years Medication Use, N (%)				
Antidiabetic	64,228 (18.93%)	102,299 (15.07%)	1.32 (1.301, 1.330)	0.0385 (0.0370, 0.0401); *p* < 0.001
Antihypertensive	189,907 (55.96%)	290,736 (42.84%)	1.70 (1.682, 1.710)	0.1312 (0.1292, 0.1333); *p* < 0.001
Antilipemic	130,012 (38.31%)	211,333 (31.14%)	1.37 (1.362, 1.385)	0.0717 (0.0698, 0.0737); *p* < 0.001
Antidepressant	204,887 (60.38%)	197,508 (29.10%)	3.71 (3.680, 3.745)	0.3128 (0.3108, 0.3147); *p* < 0.001
Anxiolytic/Sedative/Hypnotic	95,743 (28.21%)	72,202 (10.64%)	3.30 (3.2661, 3.3371)	0.1758 (0.1741, 0.1774); *p* < 0.001
Antineoplastics	4216 (1.24%)	8804 (1.30%)	0.96 (0.993, 0.993)	−0.0005 (−0.0010, −0.0001); *p* = 0.0203
Antipsychotic	181,065 (53.36%)	21,916 (3.23%)	34.28 (33.769, 34.801)	0.5013 (0.4995, 0.5030); *p* < 0.001
Opioids	168,226 (49.57%)	217,786 (32.09%)	2.08 (2.063, 2.098)	0.1748 (0.1728, 0.1769); *p* < 0.001
Antiarrhythmics	2209 (0.65%)	4482 (0.66%)	0.99 (0.937, 1.037)	−0.0001 (−0.0004, 0.0002); *p* = 0.5788
Antiretroviral	28,475 (8.39%)	34,790 (5.13%)	1.70 (1.668, 1.723)	0.0327 (0.0316, 0.0337); *p* < 0.001
Thyroid	22,685 (6.68%)	29,882 (4.40%)	1.56 (1.528, 1.583)	0.0228 (0.0218, 0.0238); *p* < 0.001
Substance Use: Alcohol or Illicit Drug	114,103 (33.62%)	74,029 (10.91%)	4.14 (4.095, 4.181)	0.2272 (0.2254, 0.2289); *p* < 0.001
Outpatient encounters	76.14 ± 106.44	29.46 ± 47.90	2.58 (2.572, 2.597)	*p* < 0.001
Inpatient admissions	2.47 ± 2.67	1.87 ± 1.80	1.32 (1.305, 1.330)	*p* < 0.001
Emergency Department visits	2.31 ± 2.75	1.71 ± 1.48	1.35 (1.341, 1.365)	*p* < 0.001
Orders for COVID-19; N (% SMI Group) with COVID-19 Order	N = 130,299 (38.40%)	N = 168,968 (24.90%)	1.88 (1.864, 1.897)	0.1350 (0.1331, 0.1369); *p* < 0.001
Sum of Orders for COVID-19; Mean SD	0.61 ± 0.99	0.33 ± 0.65	1.85 (1.842, 1.868)	*p* < 0.001
COVID-19 Test Results; N (% w COVID-19 Order) with COVID-19 Test	N = 117,998 (90.56%)	N = 149,005 (88.19%)		
Positive COVID-19 Test Result	12,190 (10.33%)	18,649 (12.52%)	0.81 (0.786, 0.825)	−0.0219 (−0.0243, −0.0194); *p* < 0.001

* Matching criteria. ** RD (95% CI); *p*-value via Chi-square for categorical predictors and generalized linear regression model with binomial distribution and negative binomial distribution for continuous predictors.

**Table 2 behavsci-14-00356-t002:** Group differences (SMI vs. non-SMI) for outpatient, inpatient, and emergency services utilization and deaths, comparing the first six months post emergency declaration (i.e., 15 March 2020 to 15 September 2020) to the same six-month period of the year pre-pandemic (i.e., 15 March 2019 to 15 September 2019).

Group Differences in Pre–Post-Pandemic Utilization * (Mean ± SD)	SMI (N = 339,349)	Non-SMI (N = 678,698)	Unadjusted OR (95% CI)	Adjusted OR (95% CI)
Outpatient visits	−1.46 ± 16.30	−0.66 ± 8.64	2.12 (2.102, 2.137)	1.50 (1.484, 1.510)
Inpatient admissions	−0.27 ± 1.63	−0.22 ± 1.39	1.18 (1.158, 1.206)	1.14 (1.118, 1.171)
Emergency room visits	−0.23 ± 1.46	−0.17 ± 1.15	1.19 (1.164, 1.208)	1.15 (1.121, 1.170)
Deaths N (%)	5894 (1.74)	4871 (0.72)	2.45 (2.354, 2.540)	0.91 (0.839, 0.990)

* Negative values indicate higher pre-pandemic utilization.

**Table 3 behavsci-14-00356-t003:** Group differences (individual SMI diagnostic categories vs. non-SMI) for outpatient, inpatient, and emergency services utilization comparing the first six months post emergency declaration (i.e., 15 March 2020 to 15 September 2020) to the same six-month period of the year pre-pandemic (i.e., 15 March 2019 to 15 September 2019).

	Group Differences in Pre−Post-Pandemic Utilization * (Mean ± SD) and OR (95% CI)		
	Outpatient Visits	Inpatient Admissions	Emergency Room Visits
Bipolar (N = 184,047)	−1.26 ± 14.72	−0.31 ± 1.59	−0.22 ± 1.38
Schizophrenia (N = 91,734)	−1.81 ± 18.15	−0.23 ± 1.65	−0.25 ± 1.60
Schizoaffective (N = 33,154)	−1.50 ± 17.68	−0.23 ± 1.71	−0.22 ± 1.53
NOS Psychosis (N = 30,414)	−1.64 ± 17.82	−0.23 ± 1.64	−0.20 ± 1.41
Non-SMI (N = 678,698)	−0.66 ± 8.64	−0.22 ± 1.39	−0.17 ± 1.15
Adjusted OR (95% CI): Regression Results for Bipolar vs. Non-SMI	0.82 (0.794, 0.850)	0.94 (0.896, 0.979)	0.93 (0.886, 0.985)
Adjusted OR (95% CI): Regression Results for Schizophrenia vs. Non-SMI	1.00 (0.968, 1.037)	1.06 (1.009, 1.104)	1.02 (0.967, 1.075)
Adjusted OR (95% CI): Regression Results for Schizoaffective vs. Non-SMI	1.04 (1.001, 1.074)	1.01 (0.969, 1.061)	0.98 (0.931, 1.041)
Adjusted OR (95% CI): Regression Results for NOS Psychosis vs. Non-SMI	0.85 (0.833, 0.871)	0.95 (0.920, 0.980)	0.92 (0.887, 0.954)

* Negative values indicate higher pre-pandemic utilization.

**Table 4 behavsci-14-00356-t004:** Proportion of patients with and without SMI, with a positive or negative COVID-19 test, who died during the observation period.

		Non-SMI	SMI	Total
Had a Negative COVID-19 test *	**Did not die**	126,604	101,246	227,850
	Total % (Row %)	53.61 (55.56)	42.87 (44.44)	96.48 (100)
	**Died**	3752	4562	8314
	Total % (Row %)	1.59 (45.13)	1.93 (54.87)	3.52 (100)
	**Total**	130,356	105,808	236,164
	**%**	55.20	44.80	100
Had a Positive COVID-19 test *	**Did not die**	17,951	11,295	29,246
	Total % (Row %)	58.21 (61.38)	36.63 (38.62)	94.83 (100)
	**Died**	698	895	1593
	Total % (Row %)	2.26 (43.82)	2.90 (56.18)	5.17 (100)
	**Total**	18,649	12,190	30,839
	**%**	60.47	39.53	100

* The estimated conditional odds ratios are 1.52 (95%CI: 1.455, 1.589) for negative COVID-19 test, and 2.04 (95% CI: 1.841, 2.256) for positive test result. Veterans without SMI had a better chance of survival than veterans with SMI for positive or negative COVID-19 test results. The odds ratio for death adjusted for COVID-19 test result is 1.59 (Mantel–Haenszel 95%CI: 1.53, 1.66).

**Table 5 behavsci-14-00356-t005:** Association between the outcome of death versus potential risk factors included in the logistic regression model (GLM). The reference groups used in this model were as follows: female for gender and White for race.

Covariates		OR (95% CI)	S.E.	*p*-Value
SMI	Bipolar	0.73 (0.654, 0.814)	0.056	<0.0001
	NOS Psychosis	1.26 (1.086, 1.458)	0.075	0.002
	Schizoaffective	0.91 (0.773, 1.075)	0.084	0.269
	Schizophrenia	1.00 (0.901, 1.108)	0.053	0.990
Age		1.07	0.002	<0.0001
Gender *	Male	1.54	0.085	<0.0001
Race **	Black	0.89	0.046	0.010
	Missing	1.21	0.127	0.139
	Other	1.05	0.129	0.698
Ethnicity	Hispanic	0.94	0.076	0.376
Number of Meds		0.95	0.013	<0.0001
CAN		1.01	0.001	<0.0001
Difference in outpatient visits		1.00	0.001	0.072
Difference in inpatient visits		1.16	0.014	<0.0001
Difference in ED visits		1.00	0.014	0.998
Tested positive for COVID-19 (Yes = 1)		1.13	0.057	0.032

* Reference = female. ** Reference = White.

## Data Availability

The datasets presented in this article are not readily available because of VA Research Information and Security Policy. Requests to access the datasets should be directed to the Pittsburgh VA Information and Technology Office.

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
