# Peer review of "The Impact of the Global Pandemic on Veterans with Serious Mental Illness (SMI): Healthcare Utilization and Mortality"

_behavsci, 2024, doi:10.3390/bs14050356_

Round 1

Reviewer 1 Report

Comments and Suggestions for Authors

The impact of the global pandemic on Veterans 

healthcare utilization and mortality

Abstract: Well, drafted. Abbreviation like SMI need to be written in full forms in the abstract and Title too.

Introduction: Well, drafted

-Patients with SMI experienced a larger decrease in outpatient, inpatient, and emergency department visits compared to non-SMI patients during the pandemic.

-The mortality rate was higher in both SMI and non-SMI individuals during the pandemic, with SMI patients being approximately 2.5 times more likely to die than non-SMI patients.

The authors are trying to say that patient with SMI had a decrease in outpatient, inpatient and emergency as compared to the non-SMI. But the mortality was higher.

(Can the authors explain this further in the discussion and reason, it would be interesting to learn)

Material & Methods:

SMI Diagnoses: the SMI sample included individuals with the following diagnoses: 82 bipolar disorder (ICD-10 codes F30-31), schizophrenia (ICD-10 codes F20, F22-24), 83 schizoaffective (ICD-10 code F25), psychosis (ICD-10 codes F28-29).

Q: Why only these psychiatry disorders included, it would be interesting to understand the authors rational for the same.

Results: Table 1-5 needs formatting, central alignment please follow standard table formatting settings.

Q Results are written well. It would also be good to format on the basis on significant and non-significant results in addition to the odds ratio.

Discussion: Needs comparisons from other studies’ findings. These needs work.

--It’s not clear if SMI was the cause of mortality or the chronic disease these patient population suffering added to the burden???

Conclusion: Needs improvement.

Overall impression: The research questions the authors trying to study is interesting, but basic things like writing the results, tables and discussion needs work. Authors must improvise to standard quality of help. (It felt like authors lost interest once reaching discussion and conclusions)

Comments on the Quality of English Language

Minor edits

Reviewer 2 Report

Comments and Suggestions for Authors

This is a helpful paper and one which identifies a small subgroup of veterans with a clear increased risk of death during the recent pandemic. In that way it may identify a group more in need of support in future pandemics. For that reason it is worthy of publication. It would have been helped if a non-veteran  group were included in the analysis. The VA is fortunate enough in having this data base to call upon, not entirely unique as the Scottish Veteran Study has similar and is reporting excellent veteran v non veteran data.

I would suggest not to use abbreviations in the title 'SMI', nor in the abstract without defining it for the reader. It would very helpful to describe and or define a 'serious' mental illness as opposed to a 'mental illness'. This would make the analysis and table 1 more clear as clearly people with non-SMI (ie still have mental illness) are in that group. I accept table 3 lists 4 diagnostic groups (including NOS) that are SMIs but were others included in SMI? A brief description appears early in the discussion but this would have helped in the introduction.

Table 1 has a slight odd anomaly in that patients with SMI are more likely to be obese but they also have a lower BMI - this requires explaining (may be very skewed weight distribution?).

In table 2, I wasnt clear if this was adjusted/censored for death? Clearly a group with a higher death rate will have a lower outpatient and potentially inpatient frequency than a group of survivors. This is an important point to make clear or present data.

The discussion makes a reasonable attempt to explain some of the statistics presented but may still be overstating the authors ideas, such as the diagnoses being NOS being early or unstable and of course not having cause of death the reader is still left unsure if the patients are dying of or with COVID given those with SMI have a significant burden of comorbidity, exactly those one would expect to die.

One last point the discussion mentions health care may be sought outside the VA system and this may also include COVID testing, especially for those more capable (possibly the ones without SMI) which would significantly affect some of the results and discussion.

Minor typos exist such as line 46 starting without a capital letter 'the' rather than 'The' a careful review is required.
